# Neuroticism and Conscientiousness Moderate the Effect of Oral Medication Beliefs on Adherence of People with Mental Illness during the Pandemic

**DOI:** 10.3390/brainsci12101315

**Published:** 2022-09-29

**Authors:** Fabio Ferretti, Arianna Goracci, Pier Francesco Laurenzi, Rossella Centola, Irene Crecchi, Aldo De Luca, Janette Monzillo, Omar Guidi, Giusy Sinigaglia, Giacomo Gualtieri, Lore Lorenzi, Alessandro Cuomo, Simone Bolognesi, Valter Travagli, Anna Coluccia, Andrea Fagiolini, Andrea Pozza

**Affiliations:** 1Department of Medical Sciences, Surgery and Neurosciences, University of Siena, 53100 Siena, Italy; 2Department of Molecular Medicine, School of Medicine, University of Siena, 53100 Siena, Italy; 3Department of Mental Health, Medical Center (AOUS), University of Siena, 53100 Siena, Italy; 4UOC Farmacia, AOU San Giovanni di Dio e Ruggi d’Aragona, 84131 Salerno, Italy; 5UOC Farmacia Ospedaliera, Azienda Ospedaliera Universitaria Senese, 53100 Siena, Italy; 6Farmacia, Istituto Nazionale Malattie Infettive “L. Spallanzani” IRCCS, 00149 Rome, Italy; 7Scuola di Specializzazione in Farmacia Ospedaliera, University of Siena, 53100 Siena, Italy; 8Psychology Unit, Department of Mental Health, Azienda Ospedaliera Universitaria Senese, 53100 Siena, Italy

**Keywords:** Big Five personality, adherence, beliefs about medications, COVID-19 pandemic, mental disorders, stress

## Abstract

Background. After the declaration of the pandemic status in several countries, the continuity of face-to-face visits in psychiatric facilities has been delayed or even interrupted to reduce viral spread. Little is known about the personality factors associated with medication beliefs and adherence amongst individuals with mental illness during the COVID-19 pandemic. This brief report describes a preliminary naturalistic longitudinal study that explored whether the Big Five personality traits prospectively moderate the effects of medication beliefs on changes in adherence during the pandemic for a group of outpatients with psychosis or bipolar disorder. Methods. Thirteen outpatients undergoing routine face-to-face follow-up assessments during the pandemic were included (41 observations overall) and completed the Revised Italian Version of the Ten-Item Personality Inventory, the Beliefs about Medicines Questionnaire, the Morisky Medication Adherence Scale—8-item and the Beck Depression Inventory—II. Results. Participants had stronger concerns about their psychiatric medications rather than beliefs about their necessity, and adherence to medications was generally low. Participants who had more necessity beliefs than concerns had better adherence to medications. People scoring higher in Conscientiousness and Neuroticism traits and more concerned about the medication side effects had poorer adherence. Conclusions. These preliminary data suggest the importance of a careful assessment of the adherence to medications amongst people with psychosis/bipolar disorder during the pandemic. Interventions aimed to improve adherence might focus on patients’ medication beliefs and their Conscientiousness and Neuroticism personality traits.

## 1. Introduction

After the declaration of the pandemic status in several countries, to reduce viral spread, the continuity of face-to-face follow-up visits in psychiatric facilities has been delayed or interrupted [1,2]. As a consequence of this change in psychiatric settings, reduced access to mental health services has become one of the predictors of poor adherence to oral medications, which in turn has increased the risk of symptom deterioration and relapse [3,4,5,6].

Regardless of the stressful effects of the pandemic, poor adherence to psychiatric medications, such as antipsychotics and mood stabilizers, is a relevant phenomenon in people with mental illness, as around 40% of people with psychosis or bipolar disorder show poor adherence, including discontinuation/intermittent intake of medication [7,8]. Such behavior can negatively impact the clinical outcomes, thus increasing the risk of recurrence of episodes [7,9,10].

Several conceptualizations have been developed to understand the factors that influence medication adherence in several medical settings, amongst the most important of which are the beliefs that patients hold about their treatment pathway and medicines [11,12,13,14,15]. According to the Necessity–Concerns Framework [16,17], these beliefs comprise the necessity of medicines in terms of potential for effectiveness but also concerns about possible adverse consequences. Based upon these beliefs, patients may decide not to continue with the prescribed medication if they perceive the necessity or the chances of success to be low or the costs (e.g., possible side effects) to be high. An increasing amount of data support the validity of this model and suggest that poor adherence is associated with beliefs of high concern and low necessity [18,19,20]. The model has received empirical support also in samples of individuals with mental illness. For example, Clatworthy and colleagues [21] found that low adherence was predicted by greater doubts about personal need for treatment and stronger concerns about potential negative effects, and that these predictors were independent of current mood state, illness, and demographic characteristics. In line with the Necessity–Concerns Framework, it has been found that medication beliefs change over time and temporal changes in medication beliefs can predict changes in adherence in the long run [22].

The Big Five model of personality [23,24] assumes that individual differences in personality characteristics can be organized into five broad trait domains: Extraversion (extraverted people experience high levels of happiness and life satisfaction), Agreeableness (individuals high in this trait are helpful, warm, and emphatic), Conscientiousness (persons high in this trait tend to be well-organized, goal-directed, and persistent), Neuroticism (persons high in this trait tend to experience strong levels of distress), and Openness (people high in openness have broad interests and seek experiences).

A relatively large number of studies have investigated the relationship between the Big Five personality traits and medication adherence in patients with chronic medical conditions [24,25]. Findings from systematic reviews showed that amongst the Big Five factors, Neuroticism and Conscientiousness are negatively and positively associated with medication adherence, respectively [26,27].

Some authors explored the role of Big Five personality traits in medication adherence amongst people with psychiatric disorders, showing that only low Conscientiousness predicted lower adherence [28]. Other researchers reported that in people with chronic diseases, Agreeableness and Conscientiousness were positively related to medication adherence, while Neuroticism was negatively associated with adherence behavior [29].

### Rationale and Objectives of the Present Study

In summary, little is known about the factors associated with medication adherence amongst individuals with mental illness during the pandemic. Since face-to-face psychiatric visits have been delayed or even interrupted during the pandemic, a clearer picture of the psychological variables related to medication adherence during this stressful period may help clinicians and policymakers to plan specific interventions aimed at increasing the adherence of patients with mental disorders, in order to improve the delivery of psychiatric care and the overall effectiveness of services in the long run, when highly stressful events similar to the pandemic occur.

Starting from the available evidence, the role of medication beliefs on adherence might be moderated by specific Big Five personality traits. This brief report describes a preliminary naturalistic study that explored whether the Big Five personality traits moderated the effects of medication beliefs on changes in adherence during the pandemic for a group of outpatients with mental illness.

## 2. Materials and Methods

### 2.1. Sample Population, Eligibility Criteria, and Design

The present study was conducted in outpatients with mental illness, i.e., psychosis or bipolar disorder, recruited at the Psychiatry Unit of the Santa Maria alle Scotte University Hospital of Siena, Italy. Participants were included if: they met the criteria for a primary diagnosis of psychotic/bipolar disorder according to the Diagnostic and Statistical Manual of Mental Disorders—fifth edition (DSM-5) [30]; they were on antipsychotics and/or mood stabilizers; they were aged 18 years or older; and they provided informed consent. Participants were excluded if they had a neurological disorder, mental disability, and/or difficulty understanding written Italian. Through a longitudinal design, each one of the included patients completed the measures (for a detailed description of the measures, see paragraph below) during every monthly/bimonthly routine face-to-face follow-up psychiatric assessment from November 2020 to April 2021.

### 2.2. Measures

The Beliefs about Medicines Questionnaire (BMQ) [31] was used to assess patients’ cognitions regarding the medications. The BMQ is an 11-item questionnaire comprising two subscales assessing beliefs about medication prescribed for a condition, relating to perceptions about the personal need for medication (Necessity subscale) and concerns about potential negative effects from medication (Concerns subscale). The following are representative items for the Necessity and Concerns subscales, respectively: “My life would be impossible without these medications”, “I sometimes worry about the long-term effects of these medications”. The Italian translation showed good reliability [32].

The Morisky Medication Adherence Scale—8-item (MMAS-8) [33,34,35] was used to measure adherence to medications. It is an 8-item questionnaire on a 5-point Likert scale that measures adherence to medications for a given condition. It showed acceptable reliability [36].

The Beck Depression Inventory—II (BDI-II) [37] was used to assess depressive symptoms. The BDI-II is a 21-item questionnaire that rates the severity of depressive symptoms. Higher scores denote higher depression. The Italian version showed excellent reliability [38].

The Revised Italian Version of the Ten-Item Personality Inventory (I-TIPI) [39] was used to assess the Big Five personality factors. The I-TIPI is a revised version of the Ten-Item Personality Inventory [40], a questionnaire consisting of 10 items on a seven-point Likert scale that evaluates personality traits through five scales corresponding to the Big Five factors. It showed good reliability [39].

### 2.3. Statistical Analyses

Owing to the longitudinal design, a generalized estimating equation (GEE) model was selected to examine the relationship between a dependent variable (medication adherence, measured by the MMAS-8) and a set of predictors (beliefs about medication, BMQ; symptoms of depression, BDI-II; personality traits, I-TIPI). The difference between the BMQ Necessity and BMQ Concerns subscales (ΔBMQ) was used as an independent variable. A number of models were estimated, including models with main effects and models with both main effects and interaction terms. The goodness of fit was assessed through the quasi-likelihood under independence model criterion (QIC) and corrected QIC (QICC), looking at the model that minimized these two indices. A post-hoc power analysis was performed according to the formula proposed by Li and McKeague [41]. Given a power of 90%, a type 1 error of 0.01, an effect size of 0.025, and a conditional marginal variance estimated at 1.24, the results provided a sample size of 13 clusters (subjects) with at least 2 measurements. The assumptions of the final model were checked, and the significance level was set at *p* < 0.05. The analyses were performed using the Statistical Packages for Social Science (SPSS) v. 25, IBM, Chicago (USA).

## 3. Results

### 3.1. Descriptive Characteristics

Thirteen outpatients undergoing routine psychiatric face-to-face follow-up were included (socio-demographic/clinical features in Table 1). These subjects received a mean of three face-to-face assessments.

The scores of the questionnaires for all the measurements are summarized in Table 2. The subscales of the BMQ showed that respondents had stronger concerns about their medication than beliefs about the necessity of these medical treatments; the difference between the two subscale scores (BMQ) was 7 (IQ range = 6), showing that patients’ perceptions about medical treatments were mainly driven by their concerns. The degree of medication adherence, measured by the MMAS-8, was quite low: its median value was 3.5 (IQ range = 2) and only one patient showed the maximum score of 6, reflecting medium adherence. A mild level of symptoms of depression on the BDI-II was found (median = 14; IQ range = 16) but, taking into account all the measurements collected at the different time points, 4/40 (10.0%) revealed severe symptoms of depression and 9/40 (22.5%) showed a moderate level of depression.

Participants’ personality traits are described through the I-TIPI score: this group of subjects was characterized by high levels of Agreeableness, Conscientiousness, and Neuroticism, but low levels of Extraversion and Openness. Concerning Extraversion and Openness, respectively, 67.5% and 97.5% of the patients were below the normative data according to age and sex. Concerning Agreeableness, Conscientiousness, and Neuroticism, respectively, 65%, 55%, and 70% of the patients were above the norms.

### 3.2. Effects of Depression, Big Five Personality Traits, and Medication Beliefs on Adherence: Main and Interaction Effects

With the aim of exploring the relationship between the dependent variable, expressed by the degree of medication adherence, and its predictors (symptoms of depression, personality traits, and beliefs about medicines), a series of GEE models were estimated, choosing the model that minimized both the QIC and QICC criteria. The final model included three predictors: the main effect of ΔBMQ, and two interaction terms (I-TIPI Conscientiousness and I-TIPI Neuroticism) with ΔBMQ. The model’s QIC and QICC were 49.7 and 45.5, respectively, and the lack of multicollinearity was confirmed by comparison between Type I and Type III sum of squares estimation. The results of the model estimation are listed in Table 3. All the parameters were significant. The positive sign of ΔBMQ showed that an increasing perception about the treatment necessity compared to concerns produced a stronger degree of medication adherence (*p* = 0.000). However, the model showed that some personality traits, such as Conscientiousness and Neuroticism, affected adherence to medications when they interacted with beliefs about medications: higher scores in neuroticism and conscientiousness resulted in a significant decrease in medication adherence (respectively, *p* = 0.001; *p* = 0.004).

A dichotomous value of I-TIPI Conscientiousness and Neuroticism was used, according to the normative data proposed by Goslin and colleagues [42], to better characterize the differences. Both personality traits showed that, when Conscientiousness and Neuroticism scored low, an increase in ΔBMQ did not change MMAS-8 scores. On the contrary, a direct relationship between beliefs about medications and adherence to medications was typical of subjects with high Conscientiousness and Neuroticism.

The interaction effects of personality traits on the relationship between beliefs about medications and adherence to medications are displayed in Figure 1.

## 4. Discussion

As for people with medical disorders [43,44], the COVID-19 pandemic has represented a severely stressful life event for individuals with psychiatric conditions. The results of the present study showed that, during the pandemic, people with mental illness have shown stronger concerns about their psychiatric medications than beliefs about the necessity of these treatments, and that adherence to medications is generally low in this population. The perceived necessity of the medications seems to have a more important role in adherence than perceived concerns: people who have more necessity beliefs than concerns would have better adherence to medications.

The present findings expand current knowledge about the role of medication beliefs on adherence, highlighting the moderating effects of specific personality traits. Interestingly, specific Big Five personality traits seem to moderate the relationship between medication beliefs and adherence; in particular, people with higher levels of Conscientiousness or Neuroticism traits and who perceived more concerns than necessity about the medication side effects seem to have poorer adherence. This appears to be partly consistent with the findings reported in samples with psychiatric or other chronic medical disorders assessed before the pandemic [27,28,29], where Neuroticism was a predictor of poorer medication adherence. The moderating role of Neuroticism might be related to the fact that it can include bodily self-focusing tendencies that could increase the perception of negative side effects, as reported in previous research [45].

The fact that high Conscientiousness moderates the effect of perceived concerns seems to be slightly inconsistent with the literature and the Big Five model, which assumes that Conscientiousness is a functional, goal-directed trait [24]. An explanation might be that if people who endorse higher concerns about the side effects of their medicines also have a Conscientiousness trait, they could have a persistent and goal-directed mental attitude towards a negative consideration of the side effects, which in turn could increase the likelihood of poor adherence. This explanation seems to be consistent with data highlighting the so-called downside of high levels of Conscientiousness for psychological wellbeing (i.e., the effect of high levels of Conscientiousness on low levels of wellbeing), as previously suggested [46,47]. In addition, high levels of Conscientiousness might be associated with excessive attention focused on bodily signals, typical of hypochondriac and/or obsessive–compulsive traits, which can generally be related to a catastrophic perception of the side effects of medications [48,49]. The role of Conscientiousness traits influencing a greater perception of the side effects appears to be in line also with data showing that high levels of this trait are related to a poorer understanding of the clinical information received from healthcare professionals [50].

Overall, these findings suggest the importance of the careful assessment of adherence to psychiatric medications amongst people with mental illness during the pandemic. Health education interventions aimed to improve patients’ awareness of treatment characteristics and adherence to medications during the pandemic should focus on the beliefs that individuals endorse about the medicines that they take [51]. In addition, people with Neuroticism and Conscientiousness traits and high concerns about medicines should be considered as a subgroup of patients potentially at risk of low adherence.

The small sample size should be considered as a strong limitation that might have influenced the statistical power of the analyses and prevented further analysis. Another issue is the fact that we did not explore the effects of other variables, such as different types of medications, phases of the pandemic, number of visits with psychiatric professionals during the pandemic, or patients’ socio-demographics, such as gender and foreign status, which should be analyzed in future research, since previous research showed that they are related to adherence or the clinical picture of psychosis or bipolar disorders [52,53,54]. Another shortcoming concerns the fact that the durations of the primary diagnosis and psychiatric treatment were not very homogenous, as they were around seven years, with a standard deviation or around six years. In addition, the self-report measures used to assess medication adherence should be integrated with other sources of information, such as instruments completed by informal caregivers and biological markers. Another aspect that deserves future consideration is whether the effect of personality traits and medication beliefs on adherence can, in turn, predict a higher risk of relapse.

## 5. Conclusions

This brief report showed that during the pandemic individuals with mental illness had stronger concerns about their oral psychiatric medications rather than beliefs about their necessity, and adherence to oral psychiatric medications was generally low. Individuals who had more necessity beliefs than concerns had better adherence to medications. People scoring higher in Conscientiousness and Neuroticism traits and more concerned about the medication side effects had poorer adherence. The present findings underline the importance of a careful assessment of the adherence to medications amongst people with these conditions during the pandemic. Interventions aimed to improve adherence might focus on patients’ medication beliefs and their Conscientiousness and Neuroticism personality traits.

## Figures and Tables

**Figure 1 brainsci-12-01315-f001:**
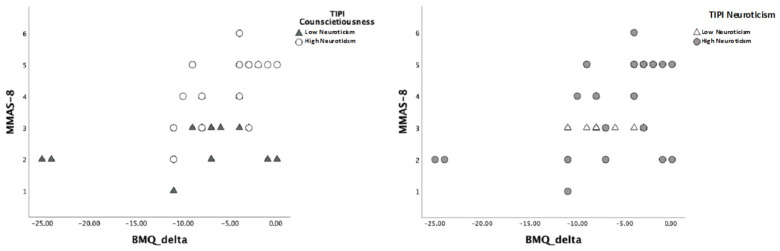
Relationship between ΔBMQ and MMAS-8 scores according to different levels of I-TIPI Conscientiousness and Neuroticism.

**Table 1 brainsci-12-01315-t001:** Socio-demographic characteristics and clinical features of the group (*n* = 13).

Age (Years) *		57.7 ± 14.4
Sex **	Female	8 (61.5)
	Male	5 (38.5)
Marital status **	Single	2 (16.7)
	Married	9 (75.0)
	Widowed	1 (8.3)
Occupational status **	Employed	4 (30.8)
	Unemployed	9 (69.2)
Education level **	High school or lower	7 (58.3)
	Degree or higher	5 (41.7)
Comorbid disorders **	Yes	5 (38.5)
	No	8 (61.5)
Type of psychiatric medications **	Antipsychotics	
	Mood stabilizers	
	Antidepressants	
Duration of primary diagnosis (months) *		85.2 ± 77.9
Duration of psychiatric treatment (months) *		93.0 ± 77.5
Previous hospitalizations **	Yes	6 (54.5)
	No	5 (45.5)
Comorbidmedical disorders **	Yes	8 (61.5)
	No	5 (38.5)

* Mean ± SD; ** *n* (%).

**Table 2 brainsci-12-01315-t002:** Descriptive statistics of the BMQ, MMAS-8, BDI-II, and I-TIPI scores.

	Mean	SD	Median	IQ Range	Min	Max
BMQ Necessity	11.4	3.8	10.5	6	5	18
BMQ Concerns	18.3	5.2	18	7	10	30
ΔBMQ (Necessity–Concerns)	−6.9	5.3	−7	6	−25	0
MMAS-8	3.5	1.2	3.0	2	1	6
BDI-II	15.7	11.9	14.0	16	0	48
I-TIPI Extraversion	3.6	1.5	3.5	2.6	1.5	6.5
I-TIPI Agreeableness	5.5	1.0	6.0	1.0	3.5	7.0
I-TIPI Conscientiousness	5.1	1.2	5.0	2.4	3.0	7.0
I-TIPI Neuroticism	5.0	1.1	5.0	1.0	2.5	7.0
I-TIPI Openness	3.5	0.8	3.5	1.4	2.0	5.0

Note. ΔBMQ was obtained by the difference between BMQ Necessity and BMQ Concerns. SD = Standard Deviation, IQ = Interquartile Range. The MMAS-8 scale, content, name, and trademarks are protected by US copyright and trademark laws. Permission for use of the scale and its coding is required. A license agreement is available from MMAR, LLC., Donald E. Morisky, ScD, ScM, MSPH, 294 Lindura Ct., USA; donald.morisky@moriskyscale.com (accessed on 5 September 2022).

**Table 3 brainsci-12-01315-t003:** Results of the model estimation.

	*β*	SE	95% CI	Wald χ^2^	*p*
Intercept	4.165	0.461	(3.262–5.068)	χ^2^_(1)_ = 81.749	0.000
ΔBMQ	0.665	0.173	(0.326–1.003)	χ^2^_(1)_ = 14.827	0.000
I-TIPI Conscientiousness × ΔBMQ	−0.065	0.023	(−0.109 to −0.021)	χ^2^_(1)_ = 8.329	0.004
I-TIPI Neuroticism × ΔBMQ	−0.049	0.014	(−0.076 to −0.021)	χ^2^_(1)_ = 11.856	0.001

## Data Availability

Data can be made available by the corresponding author upon request.

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
