# Peer review of "Neuroticism and Conscientiousness Moderate the Effect of Oral Medication Beliefs on Adherence of People with Mental Illness during the Pandemic"

_brainsci, 2022, doi:10.3390/brainsci12101315_

Round 1
Reviewer 1 Report
The article was well written. Just a few tips to improve the article.
In the introduction, more previous research is needed. Also, the research gap in this study should be explained. In other words, the importance and necessity of this study should be explained.
In the research method section, report the statistical population, the sample, the sampling method, how to choose the sample size, and the reliability and validity of the questionnaires used. Also, report the statistical test used.
In the discussion section, the results are not fully and adequately explained. Especially regarding the variable of neuroticism, the explanation should be more complete. Explain and interpret the results based on theoretical foundations and previous research.
Author Response
The article was well written. Just a few tips to improve the article.
Response: We gratefully thank the Reviewer for appreciating our work and for the precious suggesstions.
-In the introduction, more previous research is needed. Also, the research gap in this study should be explained. In other words, the importance and necessity of this study should be explained.
Response: As requested by the Reviewer, in the Introduction, we have added more previous research and new references, by adding the following sentence: “For example, Clatworthy and colleagues [15] found that low adherence was predicted by greater doubts about personal need for treatment and stronger concerns about potential negative effects, and that these predictors were independent of current mood state, illness, and demographic characteristics. In line with the Necessity-Concerns Framework, it has been found that medication beliefs change over time and temporal changes in medication beliefs can predict changes in adherence in the long run [16]”. In addition, we have discussed more in detail the importance and necessity of this study by adding a specific paragraph entitled as “1.1. Rationale and objectives of the present study”, where the following statement has been added: “Since face-to-face psychiatric visits have been delayed or even interrupted during the pandemic, a clearer picture of the psychological variables related to medication adherence during this stressful period may help clinicians and policy makers to plan specific interventions aimed at increasing the adherence of patients with mental disorders, in order to improve the delivery of psychiatric care and the overall effectiveness of services in the long run, when highly stressful events like the pandemic occur.”
-In the research method section, report the statistical population, the sample, the sampling method, how to choose the sample size, and the reliability and validity of the questionnaires used. Also, report the statistical test used.
Response: As requested by the Reviewer, we have better clarified the sample population and other information related to sampling by identifying this in a specific paragraph entitled as “Sample population and eligibility criteria” and by adding the following sentence: “The present study was conducted in outpatients with mental illness, i.e., psychosis or bipolar disorder, recruited at the Psychiatry Unit of the Santa Maria alle Scotte University Hospital of Siena, Italy.” The statistical tests have been described in detail in the paragraph entitled as “Statistical analyses”.
-In the discussion section, the results are not fully and adequately explained. Especially regarding the variable of neuroticism, the explanation should be more complete. Explain and interpret the results based on theoretical foundations and previous research.
Response: As requested by the Reviewer, we have added an interpretation of the results related to neuroticism and new references based on theoretical and empirical research, by adding the following sentence: “This appears to be partly consistent with the findings reported in samples with psychiatric or other chronic medical disorders assessed before the pandemic [21-23], where neuroticism was a predictor of poorer medication adherence. The moderating role of neuroticism might be related to the fact that it can include bodily self-focusing tendencies that could increase the perception of negative side effects, as reported in previous research [37].”
Reviewer 2 Report
From a multidisciplinary team, this brief report addresses the exacerbation during the pandemic of low medication adherence in people with mental illness from the necessity-concerns conceptualization framework (poor adherence is associated with beliefs of high concern and low necessity) of this common problem. With the present preliminary work, they aim to provide evidence that contributes to generating knowledge in the current pandemic context on the role of the Big Five personality traits on changes of adherence in a group of outpatients with mental illness (.. Agreeableness, neuroticism, and conscientiousness have been previously described as modulating concerns and medication beliefs on adherence depending on the clinical populations). The main finding is that conscientiousness and neuroticism, with high concerns about medications, were found to model adherence and therefore may allow identifying groups at risk among people with psychosis and bipolar disorder (the population studied). Several few things still need the authors' attention, as presented below in a constructive manner.
Despite the small sample size in this preliminary result, which is the main strong limitation as also declared by the authors, the results indicate a
The other limitations (contribution of other variables, phases of the pandemic, types of medications, number of visits, self-report and informant reports, biological markers, risk of relapse) are identified and stated.
However, the sex/gender issue, which is obvious to play a role, hasn’t been mentioned, nor can expectations be driven from this preliminary data in this respect (mainly because the study addresses neuroticism)
Methods
Please, start with the description of the sample population and clinical scenario (source) (it was mentioned in the goals but should be specified ) prior to the inclusion/exclusion criteria.
In the abstract, this point is better solved as it clarifies that is was a group of outpatients with psychosis or bipolar disorder.
Results
Table 1. Check spaces in the variables (comorbid disorders, psychiatric medications, comorbid medical disorders)
The degree of severity of psychosis and bipolar disorder were not very homogeneous. This should be discussed as a limitation, should’nt it?. The mean age of the population was 57.7 + 14.4. As the data of primary diagnosis and duration of treatment were 7 years but sd were 6 .
Tables 2. Since the data is reported in a specific order, with BMQ first and the other tests, please, provide the methods in coherence.
Lines 179-181 in the discussion, lines 26-28 in the abstract
The results confirmed the necessity-concerns conceptualization, that the highest the patient’s perception about the medication needs vs concerns, the higher the adherence. However, the data (Table 2) indicates that the scenario was of lower necessity than concerns. Therefore, should the direction of the interpretation of the finding should be expressed oppositely, as shown by the negative value (-6.9) given to the ABMQ?.
The title should be informative on the preliminary findings. As it stands right now does not say anything new about what could be expected and has already been reported in MH and other diseases. I’d strongly suggest giving ‘the main conclusion, which summarizes table 3, instead, for a straightforward home-take message for readers. It is unclear if the title is lines 2-3 or line 4. The latter would be the one suitable for a clear home take message.
Author Response
From a multidisciplinary team, this brief report addresses the exacerbation during the pandemic of low medication adherence in people with mental illness from the necessity-concerns conceptualization framework (poor adherence is associated with beliefs of high concern and low necessity) of this common problem. With the present preliminary work, they aim to provide evidence that contributes to generating knowledge in the current pandemic context on the role of the Big Five personality traits on changes of adherence in a group of outpatients with mental illness (.. Agreeableness, neuroticism, and conscientiousness have been previously described as modulating concerns and medication beliefs on adherence depending on the clinical populations). The main finding is that conscientiousness and neuroticism, with high concerns about medications, were found to model adherence and therefore may allow identifying groups at risk among people with psychosis and bipolar disorder (the population studied). Several few things still need the authors' attention, as presented below in a constructive manner.
Response: We gratefully thank the Reviewer for having carefully read our paper and for highlighting the strengths of our work.
-Despite the small sample size in this preliminary result, which is the main strong limitation as also declared by the authors, the results indicate a
The other limitations (contribution of other variables, phases of the pandemic, types of medications, number of visits, self-report and informant reports, biological markers, risk of relapse) are identified and stated.
However, the sex/gender issue, which is obvious to play a role, hasn’t been mentioned, nor can expectations be driven from this preliminary data in this respect (mainly because the study addresses neuroticism).
Response: As requested by the Reviewer, we have discussed the role of gender that should be explored in future research, as follows: "Another issue is the fact that we did not explore the effects of other variables, such as different types of medications, phases of the pandemic, number of visits with psychiatric professionals during the pandemic or patients’ socio-demographics such as gender and foreign status, which should be analysed in future research, since previous research showed that they are related to adherence or the clinical picture of psychosis or bipolar disorders [44-46]."
Methods
-Please, start with the description of the sample population and clinical scenario (source) (it was mentioned in the goals but should be specified) prior to the inclusion/exclusion criteria.
In the abstract, this point is better solved as it clarifies that is was a group of outpatients with psychosis or bipolar disorder.
Response: As requested by the Reviewer, we have clarified the sample population and clinical scenario prior to the inclusion/exclusion criteria, as follows: "The present study was conducted in outpatients with mental illness, i.e., psychosis or bipolar disorder, recruited at the Psychiatry Unit of the Santa Maria alle Scotte University Hospital of Siena, Italy." In addition, to better identify the sample population, the heading of the first paragraph of the Methods section has been revised accordingly, as follows: "Sample population and eligibility criteria".
Results
-Table 1. Check spaces in the variables (comorbid disorders, psychiatric medications, comorbid medical disorders).
Response: As requested by the Reviewer, we have corrected the spaces in Table 1.
-The degree of severity of psychosis and bipolar disorder were not very homogeneous. This should be discussed as a limitation, should’nt it?. The mean age of the population was 57.7 + 14.4. As the data of primary diagnosis and duration of treatment were 7 years but sd were 6 .
Response: As requested by the Reviewer, we have added the following statement to discuss this point as a limitation: "Another shortcoming concerns the fact that the duration of the primary diagnosis and psychiatric treatment were not very homogenous, as they were around seven years with a standard deviation or around six years." We did not mention severity but duration of diagnosis and treatment since we did not assess the severity of illness directly.
-Tables 2. Since the data is reported in a specific order, with BMQ first and the other tests, please, provide the methods in coherence.
Response: As requested by the Reviewer, we have revised the order in which the measures are reported in the Methods section to make it the same as the one in Table 2.
- Lines 179-181 in the discussion, lines 26-28 in the abstract
The results confirmed the necessity-concerns conceptualization, that the highest the patient’s perception about the medication needs vs concerns, the higher the adherence. However, the data (Table 2) indicates that the scenario was of lower necessity than concerns. Therefore, should the direction of the interpretation of the finding should be expressed oppositely, as shown by the negative value (-6.9) given to the ABMQ?.
Response: Tables 2 presents the descriptive statistics of the scores on the scales, e.g., the mean and standard deviations, but it does contain the results related to the inferential statistics or the relations between ΔBMQ scores and MMAS-8 scores. Instead, Table 3 reports the inferential statistics related to the effects of the ΔBMQ scores and MMAS-8 scores. The positive value of the β coefficient associated with a p-value of .000 indicated that, in line with the Necessity-Concerns conceptualization, an increasing perception about the treatment necessity compared to concerns produced a stronger degree of medication adherence, as stated at page 5 in the Results section.
-The title should be informative on the preliminary findings. As it stands right now does not say anything new about what could be expected and has already been reported in MH and other diseases. I’d strongly suggest giving ‘the main conclusion, which summarizes table 3, instead, for a straightforward home-take message for readers. It is unclear if the title is lines 2-3 or line 4. The latter would be the one suitable for a clear home take message.
Response: As requested by the Reviewer, we have revised the title to make it shorter and more informative on the preliminary findings and based upon the main conclusion. We have also removed the second title to avoid confusion, as follows: "Neuroticism and conscientiousness personality traits moderate the effect of oral medication beliefs on adherence of people with mental illness during the pandemic"
Round 2
Reviewer 2 Report
The authors have properly answered all the queries, clarified the issues, and revised the Ms accordingly.
Author Response
We warmly thank the Reviewer for appreciating our revision efforts.